# Flow Ripple Reduction in Reciprocating Pumps by Multi-Phase Rectification

**DOI:** 10.3390/s23156967

**Published:** 2023-08-05

**Authors:** Gürhan Özkayar, Zhilin Wang, Joost Lötters, Marcel Tichem, Murali Krishna Ghatkesar

**Affiliations:** 1Department of Precision and Microsystems Engineering, Delft University of Technology, 2628 CD Delft, The Netherlands; z.wang-20@tudelft.nl (Z.W.);; 2Bronkhorst High-Tech B.V., 7261 AK Ruurlo, The Netherlands

**Keywords:** micropump, reciprocating pump, flow ripple, fluidic rectification, multi-phase rectifier

## Abstract

Reciprocating piezoelectric micropumps enable miniaturization in microfluidics for lab-on-a-chip applications such as organs-on-chips (OoC). However, achieving a steady flow when using these micropumps is a significant challenge because of flow ripples in the displaced liquid, especially at low frequencies or low flow rates (<50 µL/min). Although dampers are widely used for reducing ripples in a flow, their efficiency depends on the driving frequency of the pump. Here, we investigated multi-phase rectification as an approach to minimize ripples at low flow rates by connecting piezoelectric micropumps in parallel. The efficiency in ripple reduction was evaluated with an increasing number (n) of pumps connected in parallel, each actuated by an alternating voltage waveform with a phase difference of 2π/n (called multi-phase rectification) at a chosen frequency. We introduce a fluidic ripple factor ($RF_{fl.}$), which is the ratio of the root mean square (RMS) value of the fluctuations present in the rectified output to the average fluctuation-free value of the discharge flow, as a metric to express the quality of the flow. The fluidic ripple factor was reduced by more than 90% by using three-phase rectification when compared to one-phase rectification in the 2–60 μL/min flow rate range. Analytical equations to estimate the fluidic ripple factor for a chosen number of pumps connected in parallel are presented, and we experimentally confirmed up to four pumps. The analysis shown can be used to design a frequency-independent multi-phase fluid rectifier to the desired ripple level in a flow for reciprocating pumps.

## 1. Introduction

There are mainly two ways of controlling fluid flow in microfluidics: active flow control and passive flow control. Active flow control requires an energy input from an actuator, and passive flow control does not require any actuators for fluid flow manipulation [1,2]. Active flow control is used in many applications, such as biomedical healthcare [3], food industry [4], inkjet printing [5], therapeutics [6], and soft robotics [7]. When compared to passive flow control, active flow control may provide greater control over the flow rate, a more precise flow, and closed-loop control [8,9,10]. Reciprocating pumps are one of the preferred pump types in active flow control for the delivery of gases and low-viscosity liquids (~1 mPa·s, e.g., water) [11,12]. They contain one or more pumping elements (pistons, plungers, or diaphragms) that reciprocate into and out of pumping chambers to produce the flow. The need for portable microfluidic platforms increased the interest in miniaturized reciprocating pumps, especially piezoelectric diaphragm pumps. These pumps are ideal for liquid flow rates of less than 50 µL/min and back pressures of up to 1.5 bar [13]. They do not require priming, have simple structures, and can be miniaturized and mass-produced [14].

Low-flow rates are necessary for applications such as organs-on-chips (OoC) technology, where organ functions are emulated with the aim of minimizing animal testing [15]. The fluid flow through OoC environments needs to be stable to provide an appropriate fluid shear stress level on the cells in microfluidic chambers, e.g., below a threshold of 3 Pa [16], while mimicking vascular perfusion [17,18,19,20]. The purpose is to generate a steady flow and prevent adverse effects on cells during fluid flow, which could influence the stability and viability of the cells.

Piezoelectric diaphragm pumps are promising in terms of miniaturization [12]. The working principle and the schematic representation of the pump are shown in Figure 1. The pump consists of a chamber with a piezoelectrically actuated diaphragm and check valves at the inlet and the outlet (Figure 1a) to displace the fluid in one direction. In the suction mode, the diaphragm deforms upward to allow the fluid inside the chamber while the check-valve at the inlet is open and the check-valve at the outlet is closed. In the discharge mode, the diaphragm deforms downward to displace fluid out of the chamber, while the check valve at the inlet is closed and the check valve at the outlet is open. The functionality in fluidic circuit symbols is shown in Figure 1b. The flow generated by piezoelectric diaphragm pumps pulsates due to the reciprocating motion of the displacing diaphragm (Figure 1c). The amplitude and frequency of the diaphragm are the main parameters for flow rate control.

There are several methods reported in the literature to remove the pulsations in reciprocating pumps [21]. These methods include pump design modification by changing chamber outlet geometry [22,23], pulsation dampers, such as flexible tubing [24] and compliant membranes [25,26], controllable flow restriction by throttling with an orifice [27,28], and the multiplication of pistons [21,29,30]. Pump design modifications and dampers are often limited in their effectiveness and introduce compromises with regard to system size and response time [31]. In some cases, high pressures above 1 bar are required for dampers to be effective [32,33]. Furthermore, dampers cover only a narrow frequency range of a design and, thus, cannot reduce frequency ripples effectively over the entire flow rate range of the pump [34]. Controllable flow restriction is carried out by adding an extra proportional valve, which might produce intolerable high back pressures by increasing the resistance of the microfluidic system.

We propose that multi-phase rectification is a promising method to reduce the pulsations in reciprocating pumps. To the best of our knowledge, no research has been conducted to explore and quantify multi-phase rectification by introducing fluidic ripple factors, albeit there are a few two-phase rectification studies that reached improvements of only 35% [35] and 55% [36] for flow ripple reductions (having out-of-phase differences between pumping chambers).

In this study, we explore generalized multi-phase rectification schemes and their effect on flow rate and flow ripples. We give a theoretical analysis on multi-phase rectification; then, the obtained results were compared to experiments using off-the-shelf piezoelectric diaphragm pumps. The ripple factor value of the flow was used as a figure of merit for a steady flow with increasing numbers of parallel-phased micropumps. As a final step, we analyze and discuss the back pressure values and the influence of flow restriction.

## 2. Theory

This section gives analytical equations for the computation of fluid flow and flow ripples. The equations are based on the analogy of rectification in electrical circuits.

### 2.1. Rectification Analogy with Electrical Elements

The electrical current (I) flowing through an electrical resistor (R) for an applied voltage (V) is given by Ohm’s law (*V = I × R*). Similarly, the fluid flow ( Q) through a resistive flow channel ($R_{h}$) for an applied pressure difference (ΔP) between the inlet and the outlet is given by Hagen-Poiseuille’s law ($\Delta P=Q\times R_{h}$) [37]. This analogy is quite useful in designing purely resistive microfluidic circuits.

The process of converting alternating current (AC) to direct current (DC) is known as electrical rectification. Input current ($I_{AC}$) and average rectified output DC current ($I_{DC}$) over a time period of ω=2π are given as [38]
(1)$$I_{AC}=I_{max}sin\left(\omega t\right)$$
(2)$$I_{DC}=\frac{1}{2\pi }\int_{0}^{2\pi }I_{max}sin\left(\omega t\right)d\left(\omega t\right)\;$$

Since the π to 2π cycle of a 2π period of $I_{DC}$ is equal to zero for a half-wave rectifier, Equation (2) can be rewritten as
(3)$$I_{DC}=\frac{1}{2\pi }\int_{0}^{\pi }I_{max}sin\left(\omega t\right)d\left(\omega t\right)$$

Similarly, the root mean square value ($I_{RMS}$) is defined as
(4)$$I_{RMS}=\sqrt{\frac{1}{2\pi }\int_{0}^{\pi }{\left(I_{max}\sin\left(\omega t\right)\right)}^{2}d\left(\omega t\right)}$$

The ripple factor (RF) is used to measure the ripple content in the rectified signal. The ripple factor is defined as the ratio of the root mean square (RMS) value of the AC component present in the rectified output to the average DC component of the rectified output signal when the load is purely resistive [39]:(5)$$RF=\frac{\sqrt{I_{RMS}^{2}-I_{DC}^{2}}}{I_{DC}}=\sqrt{{\left(\frac{I_{RMS}}{I_{DC}}\right)}^{2}-1}$$

With the analogy of electrical current (I) in the electrical domain-to-flow rate (Q) in the fluidic domain, the following equation can be written for the flow rate in microfluidic rectifiers (with flow discharge only for 0 to π cycle of a 2π period in a single pump or the equivalent half-wave rectifier):(6)$$Q_{DC}=\frac{1}{2\pi }\int_{0}^{\pi }Q_{Stroke}sin\left(\omega t\right)d\left(\omega t\right)$$
(7)$$Q_{RMS}=\sqrt{\frac{1}{2\pi }\int_{0}^{\pi }{\left(Q_{Stroke}sin\left(\omega t\right)\right)}^{2}d\left(\omega t\right)}$$
(8)$$RF_{fl.}=\sqrt{{\left(\frac{Q_{RMS}}{Q_{DC}}\right)}^{2}-1}$$
where $Q_{Stroke}$ is the peak flow rate generated by one stroke of the actuator in the piezoelectric diaphragm pump. Here, valves and actuators are considered to be ideal for the sake of simplicity, i.e., valves have zero forward pressure drop, actuators have zero resistance, and no reverse flow or leakage occurs in the pumping system.

### 2.2. Multi-Phase Rectification

In multi-phase rectification, *n* micropumps are connected in parallel, as shown in Figure 2. Each individual pump is considered similar to the others in terms of the chamber, valve, and channel parameters, resulting in the same performance. The phase difference of actuation for each pump is 2π/n. A pictorial representation of the same is given in Table 1 with examples of up to five phase rectification.

For a period of T=2π and an *n*-phase rectifier configuration, there will be *n*-pumps, with each pump being actuated at a phase difference ($\varphi_{i}$). Note that each pump discharges flow only in one half-cycle (00 to π) of the actuating full-cycle waveform:(9)$$\varphi_{i}=\left(i-1\right)\times 2\pi /n,\;\;\;\;i=1,2,3,\ldots ,n$$
(10)$$Q_{DC\left(n\right)}=\sum_{i=1}^{i=n}\left(\frac{1}{2\pi }\int_{\varphi_{i}}^{\varphi_{i}+\pi }Q_{Stroke}sin\left(\omega t+\varphi_{i}\right)d\left(\omega t\right)\right)$$
(11)$$Q_{RMS\left(n\right)}=\sqrt{\sum_{i=1}^{i=n}\left(\frac{1}{2\pi }\int_{\varphi_{i}}^{\varphi_{i}+\pi }{\left(Q_{Stroke}sin\left(\omega t+\varphi_{i}\right)\right)}^{2}d\left(\omega t\right)\right)}$$
(12)$$RF_{fl.}=\sqrt{{\left(\frac{Q_{rms\left(n\right)}}{Q_{DC\left(n\right)}}\right)}^{2}-1}$$

The theoretically calculated values for $Q_{DC}$, $Q_{RMS}$, and $RF_{fl.}$ for up to nine-phase rectification with the pumps connected in parallel are shown in Figure 3. The $Q_{DC}$ and $Q_{RMS}$ values increase linearly with increasing numbers of phase-pump rectifiers because, with an increasing number of pumps, the number of pumps contributing to the effective output flow at a given phase also increases (see Table 1). The $Q_{DC-max}$ to $Q_{DC-min}$ values increase after every odd number of phase-pump rectifiers because an additional phase adds to the $Q_{DC-min}$ minimum value. The $RF_{fl.}$ value already drastically decreases by 96.6% for the three-phase rectifier, and the subsequent improvement is asymptotically low. For a nine-phase rectifier, the $RF_{fl.}$ value is 99.6% lower than that of the one-phase rectifier. Note that with the increase in the pumps, the amount of power needed adds up. A MATLAB code for calculating n pumps in parallel and a case study of up to three-phase rectification are given in Appendix D and Appendix E, respectively.

The theoretically calculated flow rates and ripple factors for up to 100-phase rectifiers (using 100 pumps connected in parallel) are given in Table 2. It is important to highlight that the ripple factor of a rectifier having an odd number of pumps (after three pumps) is less than the rectifier with a neighboring even number of pumps, and it is the same as that of the rectifier with double the number of pumps. For example, the three-phase rectifier (*n* = 3) has a smaller ripple factor than the four-phase rectifier (*n* = 4), and it has the same ripple factor as the six-phase rectifier (*n* = 6). This relationship can be seen for each odd number of phases in multi-phase rectifiers. This is because of the way the fluid discharge profiles of each individual pump overlap with respect to the applied phase difference, as shown in Table 1 and Appendix C. The effective discharge profiles of each even-numbered phase rectifier (*n* = 2,4,6,…) have cycloidal repetitions equal to the applied phase difference. For odd-numbered rectifiers (*n* = 3,5,7,…), the cycloidal repetitions are equal to half of the applied phase differences. It is advantageous to operate odd numbers of micropumps in parallel while realizing multi-phase rectification.

## 3. Materials and Methods

The following components were used for testing: micropumps (mp6, Bartels Mikrotechnik GmbH, Dortmund, Germany), an arbitrary waveform generator in one-phase rectification case (RS PRO, RSDG 805, Northants, UK), and a microcontroller in a multi-phase rectification case to control the driving frequencies and phases (Nano Every, Arduino), with amplifiers having 100× gain (BD300, PiezoDrive, Shortland, Australia), a flow sensor for monitoring the flow rate data (MFS3, Elveflow, Paris, France), a microfluidic chip as a resistive load (Fluidic 155, microfluidic ChipShop GmbH, Jena, Germany), and a flow restrictor to test the low flow rates (1/4–28 Micro-Metering Valve, IDEX, Sulzberg, Germany). The mp6 micropumps have two piezoelectric chambers connected in series with the appropriate check valves typically operated 180° out of phase. For our experiments, we used only one piezoelectric chamber and left the other chamber unconnected. The hydraulic resistance of the tubing and the flow rate sensor (≈1 × 10^3^ Pa·s/m^3^ and ≈3.4 × 10^10^ Pa·s/m^3^, respectively) are considered negligible when compared to the microfluidic chip’s hydraulic resistance (≈4.3 × 10^11^ Pa·s/m^3^). The data on flow rate over time was collected using the Elveflow Smart Interface (ESI) in a computer. The different tubings used in the experiments are (1) PTFE (Teflon) tubing (1 mm inner diameter and 100 cm long for general connections). (2) Silicone tubing (1.3 mm inner diameter and 10 cm long for microfluidic chip connections). (3) Silicone tubing (1.3 mm inner diameter and 10 cm long for each micropump).

An experimental setup was prepared to test the multi-phase rectification schemes, as shown in Figure 4. The electrical control of the micropumps was realized by writing a sinusoidal signal generation code (with phase shifts) in Arduino IDE to create a frequency range of 0.25 to 10 Hz. In the one-phase rectification experiments, a frequency range of 0.1 Hz to 2 kHz was obtained by using the arbitrary waveform generator. The driving voltage was controlled by external potentiometers. Portable voltage amplifiers having two outputs were used for realizing high amplitudes for the piezoelectric micropump actuation.

## 4. Results and Discussion

### 4.1. Performance of a One-Phase Rectifier

A one-phase rectifier was investigated by pumping deionized water. The study was carried out by activating only one pump with only one of the piezo chambers of an mp6 Bartels pump activated and leaving the other pump chamber unconnected, effectively using the pump as a one-phase rectifier. The one-phase rectifier was tested for flow over time at various frequencies and three different amplitudes (Figure 5a). Notably, the flow rate ripples are very high at low frequencies, with an increasing trend in the range of 0.1–5 Hz, followed by a decreasing trend between 5 Hz and 100 Hz. Beyond 100 Hz, low ripples were observed in the flow. Although an increase in the driving voltage increases the flow ripples, high-driving voltages are able to cover wider OoC flow rate ranges when compared to low-driving voltages. Hence, the driving amplitude was fixed to 100 V to compare single- and multi-phase rectification by covering the 2–60 µL/min flow rate range when conducting the other experiments in this study.

The average liquid flow rate derived from Figure 5a is plotted with frequency in Figure 5b, with the standard deviation shown as the error bars. The amount of power consumed for different frequencies is also plotted in the same graph. The average flow showed an asymmetrical top hat-like distribution with frequency. This means that low flow rates can also be obtained by increasing the frequency beyond (in this case) 100 Hz. However, the power consumption steeply increases with increasing frequency.

In dynamic applications of piezo actuators, the power consumption of the actuator increases linearly with the actuator’s capacitance, driving voltage, and driving frequency. The following equations are used to calculate the power consumption of the piezo actuator [40,41]:(13)$$I_{avg}=\;C*V_{pp}*f$$
(14)$$P_{pump}=\;\pi *C*V_{pp}{}^{2}*f$$
where C is the capacitance of the actuator, $V_{pp}$ is the peak–peak driving voltage, and f is the driving frequency. The total current and total power consumption of the system can be calculated by using
(15)$$I_{total}=\;I_{amplifier}+I_{pump}=I_{amplifier}+\;C*V_{pp}*f$$
(16)$$P_{total}=\;P_{amplifier}+P_{pump}$$

The individual pump’s piezo actuator capacitance is defined by the manufacturer as 16 nF [42]. At 100 V_pp_ and 1 Hz for driving voltage and frequency, an individual pump needs a 1.6 µA average current (≈3.6 µA I_RMS_) for continuous operation with a sine wave. Increasing the driving frequency of the pump results in an increase in the flowing current (see Equation 13). For example, for a driving frequency of 100 Hz with the same driving voltage, the average current flow is 0.16 mA (≈0.36 mA I_RMS_), and for 1 kHz, it is 1.6 mA (≈3.6 mA I_RMS_). The calculated I_RMS_ values are added to the 50 mA of current flowing through the amplifier, and then the I_total_ values are obtained. The system power consumption is calculated from the voltage (24 V) and current (50 mA) values of the amplifier, resulting in 1.2 W until 300 Hz, with the increase in the pump’s power consumption (with frequency) being negligible. However, the power consumption increased exponentially above 300 Hz (Figure 5b).

As seen in Figure 5, a lower frequency range (0.1–75 Hz) has high fluidic ripples when compared to a higher frequency range. Therefore, working at higher driving frequencies is advantageous to achieve low ripple factors, but at the cost of increased system power consumption [43] and decreased efficiency in terms of the check valves [44]. As a result, the flow rate starts to drop significantly.

### 4.2. Performances of Multi-Phase Rectifiers

#### 4.2.1. Improvement in Ripple Factor

We show here that multi-phase rectification will allow low-power, ripple-free low flow rates. The results of the one-phase, two-phase, and three-phase rectifiers tested at an amplitude of 100 V and varying frequencies from 0.25 to 10 Hz are shown in Figure 6. The fluidic ripple factors and the percentage improvement of the ripples from one-phase rectifier are shown in Figure 7. The experimental results of the RF_fl._ values are plotted along with the theoretical values in Figure 8. An increase in the average flow rate can be seen with an increase in the number of phase rectifications, as predicted by the theory (Figure 3). By using three-phase rectification, i.e., three micropumps in parallel with 120° phase shifts, an average improvement of 90.5% in ripple reduction was obtained for 0.25 to 10 Hz. This improvement is nearly independent of the driving frequency in the tested frequency range. When compared to the theoretical improvement of 96.6% in Figure 3 and Table 2, there is a ~6% difference. The possible reason for this difference could be due to not having exactly identical pumps from the manufacturer. Each micropump has its own tolerance for the pumping membrane and valve thicknesses, thus affecting the stiffness of the pumps. In the theoretical study, the flow profiles are ideal sinusoids since the transfer function (H(s) = pressure/flow rate) is assumed as 1 when the flow profiles are obtained. On the other hand, we observe that the flow profiles are more like distorted sinusoids in the experiments. Flow profiles are affected by at least three parameters that are neglected in the theoretical study: the threshold pressure needed to open the check valves (Figure 1), fluid inertia, and the compliance of the silicone tubings. These parameters influence the shape of the flow profiles so as to create distorted sinusoidal waveforms when compared to the theoretically described sinusoidal waveform (Table 3 and Table A3 in Appendix F). The threshold pressure required for opening the valves and fluid inertia effects are observed at the accelerating part of the flow profile, creating a delay when compared to the theoretically obtained flow profiles. Threshold pressure depends on the valve design [45,46], and it has a similar effect on a conducting fluid as threshold voltage does on diodes [47]. Fluid inertia influences the motion of the valves by adding mass to the system [48,49] and creates a delay at the accelerating part of the flow profile. The capacitive behavior of flexible tubing creates compliance in the microfluidic system, which has already been studied in microfluidics [24,50,51]. Although the capacitive effect has a major influence on flow profiles and flow ripple values, we observed that its influence on flow ripple improvements is limited to an average of ~6% in the frequency range of 0.25 to 10 Hz. The delay in the descending part of the flow profile can be up to 1 s, affecting the time constant of the output flow profile (Table 3). Due to the capacitive effect, flow rates do not reach a zero level in multi-phase rectifiers, even at very low frequencies.

Figure 5a and Figure 6 show that the flow profiles of the one-phase rectifier return to zero below 1 Hz. This indicates a fluidic time constant of approximately 1 s in our experimental setup. The flow profiles reach zero for excitation frequencies lower than 1 Hz (1 s fluidic time constant). The flow profiles do not reach zero for excitation frequencies higher than 1 Hz. For very high excitation frequencies, the flow profiles do not reach zero; the fluidic system is too slow to catch up, and, hence, the flow rate is smoothed out. Therefore, we expect the phase rectification method to be very effective below 1 Hz (above the fluidic time constant of the system). Figure 6 and Figure 7a,b clearly show that rectification is effective at least up to 10 Hz. This indicates rectification is effective until it matches the smoothness obtained by high excitation frequencies. The fluidic time constant of the system is due to compliant elements like pump membrane capacitance and silicone tubings. If non-compliant tubes are used, e.g., stainless steel tubes, the fluidic time constant will be shorter. The close-up flow profiles, the calculated fluidic ripple factors, and the flow oscillations are shown and compared with the theoretical flow profiles for 0.25 Hz in Table 3. These profiles are also merged and compared in Appendix F to analyze the deviations for each rectifier. Similar results for the 0.5 and 1 Hz frequencies are also shown in Appendix G and Appendix H.

#### 4.2.2. Influence of Multi-Phase Rectification on Back Pressure

There will be a back pressure exerted on the micropumps due to the flow resistance in the microfluidic circuit, thus influencing the pumping flow rate. Therefore, we studied the influence of back pressure on the micropump. In order to generate back pressure in the micropumps, the height of the outlet reservoir was raised above the inlet reservoir level (see Figure 4), thus increasing the hydrostatic back pressure. The average flow rate, Q, was measured as a function of the back pressure, P, for three types of rectifiers at 1 Hz, as per Figure 9. During the experiment, the microfluidic flow resistance was kept constant, and the height of the outlet reservoir (along with the microfluidic chip used as fluidic load resistance) was increased in steps of 100 mm to observe the effect of increasing hydrostatic back pressure on the average flow rate. The average flow rate linearly decreased with increasing back pressure, as expected. The one-phase and two-phase rectifiers showed similar behavior and could withstand a back pressure of about 40 mbar when the flow rate became zero. In contrast, the three-phase rectifier could withstand higher back pressure: up to 60 mbar. For a given flow rate, three-phase rectification can handle higher back pressure (P_3_ >> P_1_ & P_2_), as shown in Figure 9. This is due to the distributed pumping load among three parallel pumps, as shown by the overlapping phases (Table 1). As the rectification level increases, the phase overlapping also increases, indicating the back pressure distributed over the pumps connected in parallel. Therefore, the back-pressure capability of the system increases with increasing fluidic rectification. The influence of increasing frequency on back pressure for rectification is shown in Appendix I. As the average flow rate of the micropump increases with frequency (Figure 6), the higher the micropump frequency, the greater the amount of back pressure the given rectifier can withstand.

#### 4.2.3. Obtaining Very Low Flow Rates

Changing the height of the reservoirs to obtain very low flow rates (Figure 9) is not a practical solution; therefore, a micro-metering valve (a flow restrictor) was used after the micropumps. The micro-metering valve limits the cross-sectional area of the flow channel, leading to an increase in the back pressure and a decrease in the flow rate. The height of the inlet reservoir, micropumps, micro-metering valve, and outlet reservoir were kept at the same level. The micropump actuation was fixed at a driving voltage of 100 V and a frequency of 1 Hz. The flow rates (2, 5, and 15 µL/min) obtained by adjusting the micro-metering valve are shown in Figure 10a, and their corresponding ripple factors are shown in Figure 10b. At a flow rate of 15 µL/min, the ripple factor improvement for the three-phase rectifier is about 93% when compared to the one-phase rectifier. Although this improvement decreases to 82% and 50% at 5 and 2 µL/min flow rates, respectively, the reduction in flow ripples by the three-phase rectification is still significant. However, at very low flow rates, the micropump parts, such as the passive check valves, might become damaged due to increased back pressure. A bypass channel after the micropump connected to the inlet reservoir could minimize the damage to micropumps.

## 5. Conclusions

The ripples in a flowing fluid generated by active reciprocating pumps, like piezoelectric micropumps, can be eliminated by multi-phase rectification. More than 90% of flow ripples are eliminated by connecting three pumps in parallel and operating them at a 120° phase difference. Connecting n pumps in parallel and actuating them at a phase difference of 360°/n will asymptotically minimize the ripple factors in the fluid flow. Experiments were performed using off-the-shelf commercial piezoelectric micropumps operating in one-phase, two-phase (180°), and three-phase (120°), and between an average flow rate of 2 µL/min and 60 µL/min. A single reciprocating pump actuated at higher frequencies could also eliminate ripples in the flow, but this results in a higher flow rate or higher power consumption. Ripples can be eliminated at a lower flow rate with a single pump by actuating at a low voltage and a high frequency. But this results in less control over the back pressure. Connecting pumps in parallel and operating them at a phase difference eliminates ripples independent of the pump actuating frequency, increases the effective flow rate, and enables the pump system to withstand higher back pressure. An odd number of pumps is more effective at eliminating ripples than an even number of parallel pumps. Every additional pump used consumes an extra voltage amplifier power. Therefore, a choice has to be made on how many micropumps to use based on the allowed flow ripple factor and the power consumption. If performance, cost, and complexity are to be balanced, then a three-phase rectifier is recommended. To conclude, multi-phase fluid rectification opens an opportunity of using reciprocating pumps for a ripple-free fluid flow, independent of actuating frequency, especially for the low flow rates (<50 µL/min) needed for many organs-on-chip applications.

## Figures and Tables

**Figure 1 sensors-23-06967-f001:**
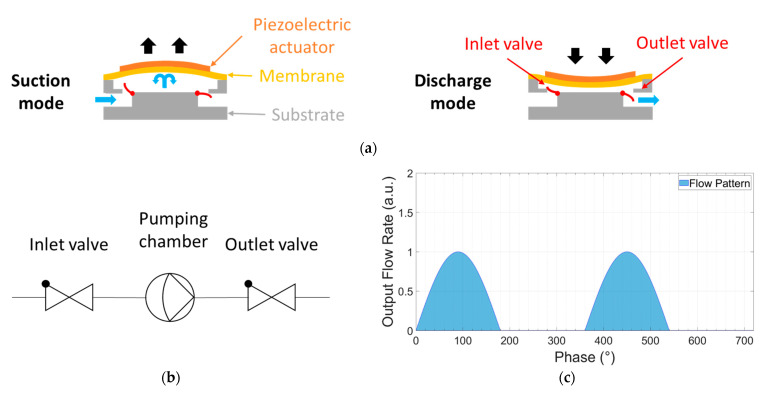
Working principle of a piezoelectric diaphragm pump. (**a**) Reciprocating motion of a piezoelectric diaphragm pump (suction and discharge modes) having two passive check valves at the inlet and outlet. (**b**) Fluidic circuit symbol representation of the micropump (as a one-phase rectifier; the electrical equivalent is shown in Appendix B), and (**c**) output flow rate characteristics with phase.

**Figure 2 sensors-23-06967-f002:**
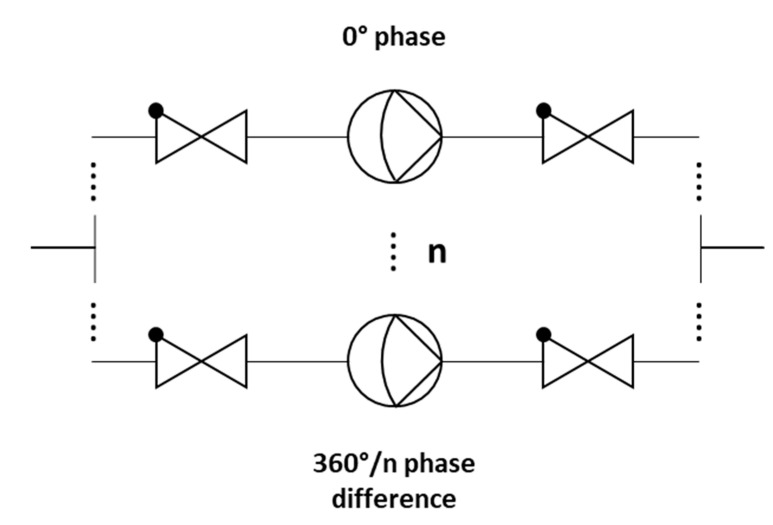
Schematic representation of a multi-phase rectifier (n micropumps in parallel with a phase shift), having check valves at the inlets and outlets.

**Figure 3 sensors-23-06967-f003:**
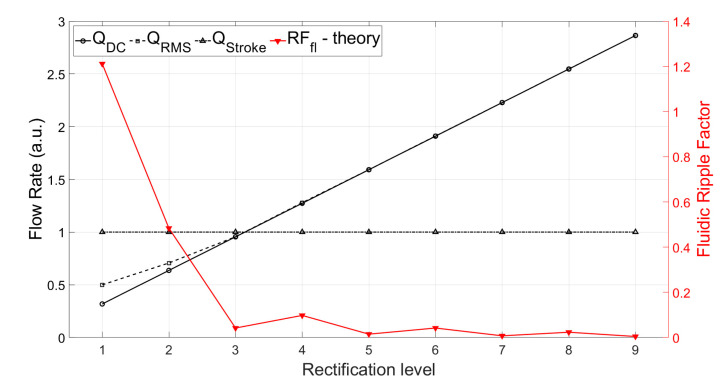
The theoretically calculated values of $Q_{DC}$, $Q_{RMS}$, and fluidic ripple factor ($RF_{fl.}$) for up to nine-phase rectifiers are shown for a chosen $Q_{Stroke}$ = 1 for every pump. The $Q_{DC}$ and $Q_{RMS}$ values increase as the phase rectification increases. For a three-phase rectifier, the ripple factor value, $RF_{fl.}$, dramatically decreases to 3.4% of that of the one-phase rectifier. The $RF_{fl.}$ value goes to 0.38% for a nine-phase rectifier. The $RF_{fl.}$ values for higher-phase rectifiers are given in Table 2 or can be calculated for any chosen number of phase rectifiers using the MATLAB code given in the Appendix A.

**Figure 4 sensors-23-06967-f004:**
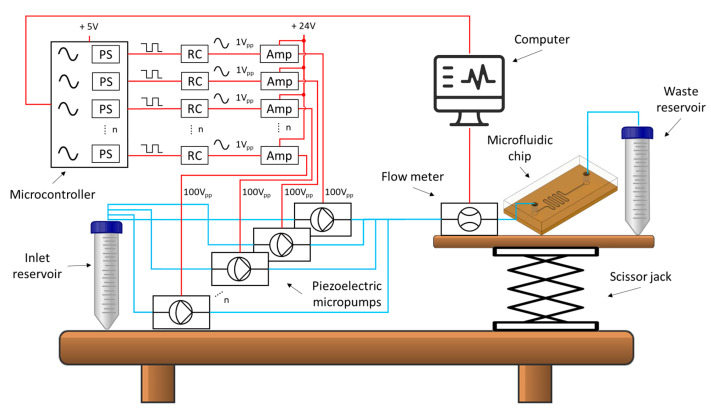
Schematic of an experimental setup for multi-phase flow rectification. The inlet and outlet reservoirs were maintained at the same height for conducting the experiments. All pumps were kept at the same height as the inlet reservoir. The height of the outlet reservoir was raised only for the back-pressure experiments (the test setup is given in Appendix E). The sinusoidal signals and their phase shifts (PS) were generated in Arduino, shaped by a resistor-capacitor (RC) circuit, and amplified (Amp) 100× before connecting them to the micropumps. The setup was tested for one-phase, two-phase, three-phase, and four-phase rectifiers, respectively. The electrical connections of the components are shown in red, and the fluidic connections are shown in blue.

**Figure 5 sensors-23-06967-f005:**
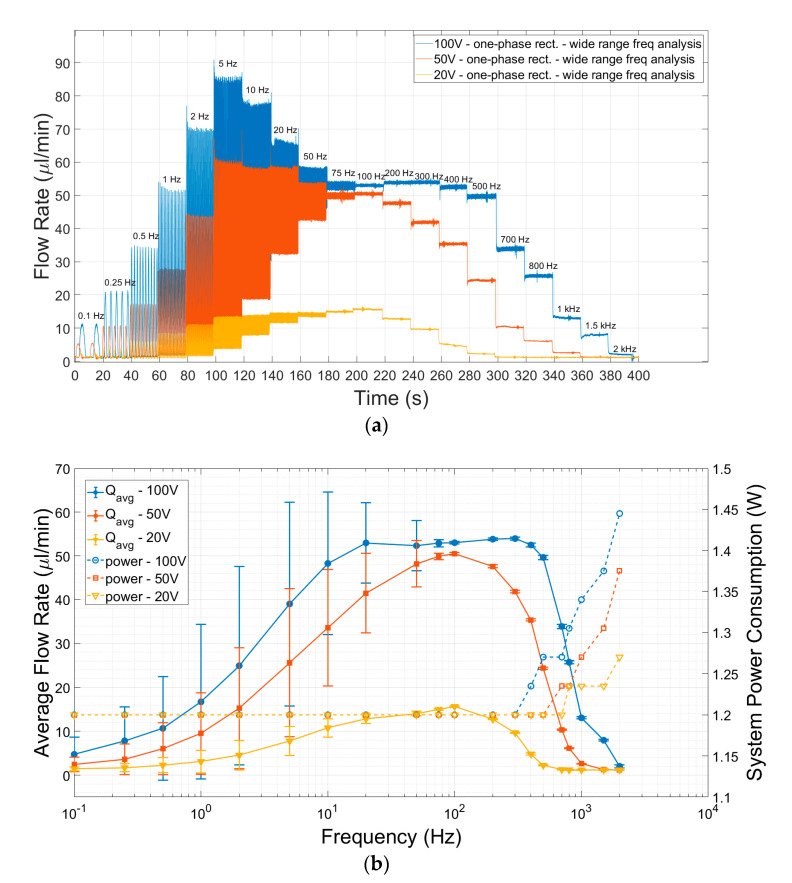
The performance of a one-phase rectifier with increasing driving frequency for three different driving voltages is shown. (**a**) The flow rate values measured over time with increasing driving frequency for three different driving actuation amplitudes of a single-chamber mp6 micropump are plotted. The ripples are large at low frequencies when compared to high frequencies. Moreover, the acquired flow rates were also reduced by increasing the frequency. (**b**) The average flow rates and the corresponding electrical power consumption with the driving frequency of the pump for three operating voltages from the data of Figure 5a are plotted. The error bars represent the standard deviation from the average flow rate. Low ripple factor flow rates can also be obtained by increasing the frequency of the micropump, but electrical power consumption increases exponentially above 300 Hz for 100 V of driving amplitude. Note that frequency (*x*-axis) is shown on a logarithmic scale.

**Figure 6 sensors-23-06967-f006:**
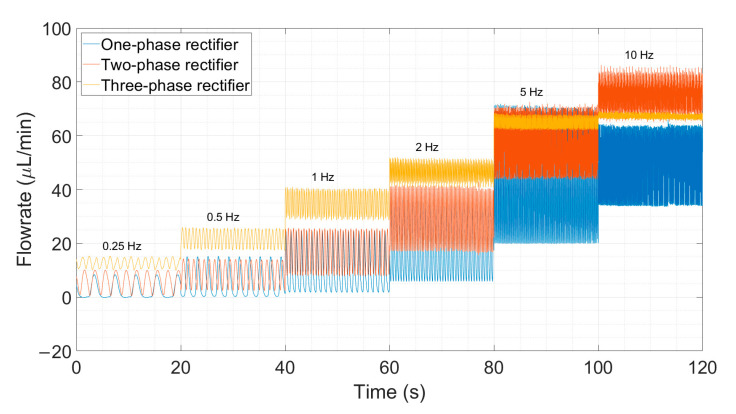
The influence of rectification on ripple reduction from one-phase to three-phases at a lower frequency range (0.25 to 10 Hz) is shown. The measurement durations are 20 s for each frequency. A one-phase rectifier uses one chamber of the mp6 pump, the two-phase rectifier uses a single chamber from two mp6 pumps connected in parallel, and similarly, a three-phase rectifier uses three mp6 pumps. Note that the flow rate also increases with the rectification level, as predicted and shown in Figure 3.

**Figure 7 sensors-23-06967-f007:**
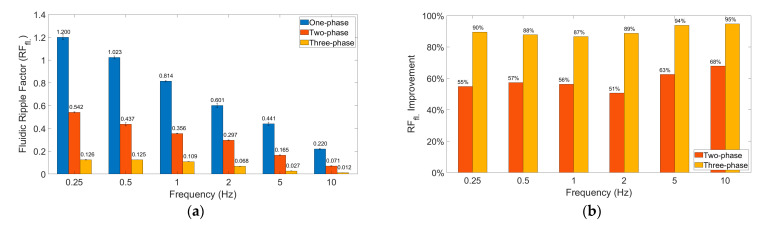
The fluidic ripple factors and ripple factor improvement with increasing phase rectification are calculated from the data in Figure 6 and were analyzed. (**a**) Fluidic ripple factors of one-, two-, three-, and four-phase rectifiers, respectively, are shown. (**b**) The percentage improvement of the fluidic ripple factors ($RF_{fl}$_._) of the two-phase and three-phase rectifiers compared to the one-phase rectifier is shown. Three-phase rectification has the lowest ripples and an average improvement of 90.5% when compared to one-phase rectification in the frequency range 0.25–10 Hz.

**Figure 8 sensors-23-06967-f008:**
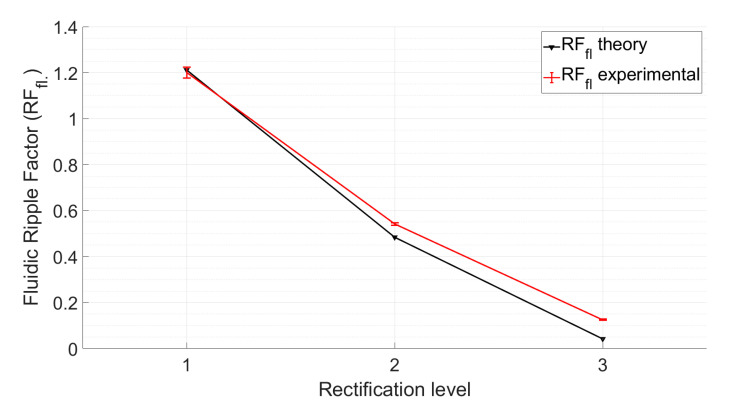
Comparison of the theoretical fluidic ripple factor and experimental average fluidic ripple factor measurements for up to a three-phase rectifier, with configurations of 100 V and 0.25 Hz. The error bars are the standard deviation obtained from five repeated experiments. It should be noted that the theoretical model does not include parameters, such as fluidic resistance, fluidic capacitance, pump membrane compliance, piezo hysteresis, and valve behavior, in the pump.

**Figure 9 sensors-23-06967-f009:**
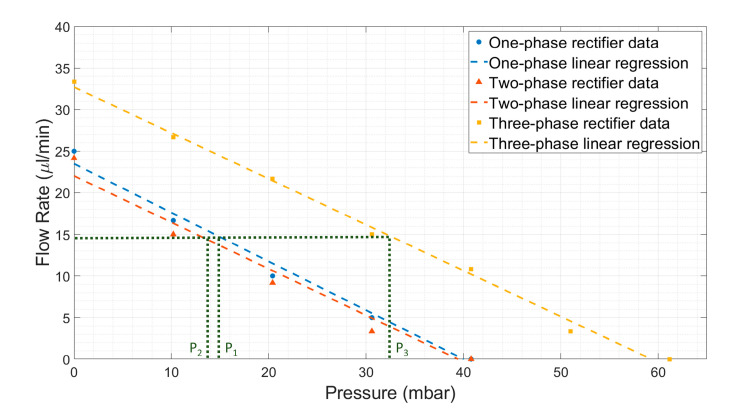
The average flow rate with back pressure for one-phase, two-phase, and three-phase rectifier mechanisms with micropump(s) actuated at 100 V and 1 Hz operating conditions. The dotted line indicates that for a given average flow rate, the three-phase rectifier can withstand higher back pressure (P_3_) compared to two-phase (P_2_) and one-phase (P_1_) rectifiers. The values of P_1_ and P_2_ are lower than P_3_ and are close together because there is no phase overlap for one-phase or two-phase rectifiers, whereas, for three-phase rectifiers, there is either a phase overlap indicating a back-pressure load distribution between pumps or a higher piezo stroke (see Table 1).

**Figure 10 sensors-23-06967-f010:**
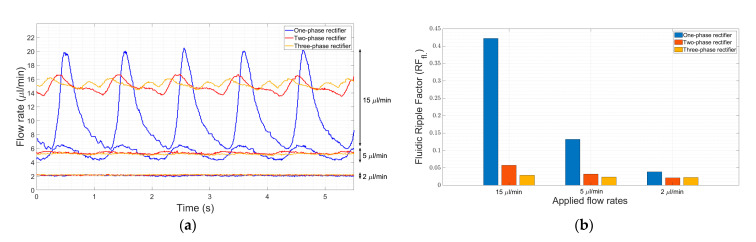
The effect of the micro-metering valve for different rectifiers at very low flow rates. (**a**) A comparison of flow rate ripples for one-phase, two-phase, and three-phase rectifiers at low flow rates. (**b**) The fluidic ripple factor values of one-, two-, and three-phase rectifiers compared to the one-phase rectifier.

**Table 1 sensors-23-06967-t001:** The effect of rectification on the output flow rate for multi-phase rectification for pumps connected in parallel for up to n = 5, five-phase rectification (flow profiles and descriptions from n = 6 (six-phase) to n = 9 (nine-phase) rectification are given in Appendix C). Each sinusoidal peak is the result of a piezoelectric stroke of each micropump (Figure 1). The different colors represent the flow profiles of different pumps. With an increase in the phase rectification (number of pumps), the strokes produced by pumps overlap, thus effectively increasing the $Q_{DC}$ with the number of pumps. The amount of overlap of the strokes also influences the $Q_{DC-max}-Q_{DC-min}$ value: lower for an odd-phase rectifier between two adjacent, even-phase rectifiers.

Rectified Output Flow Profiles	Description
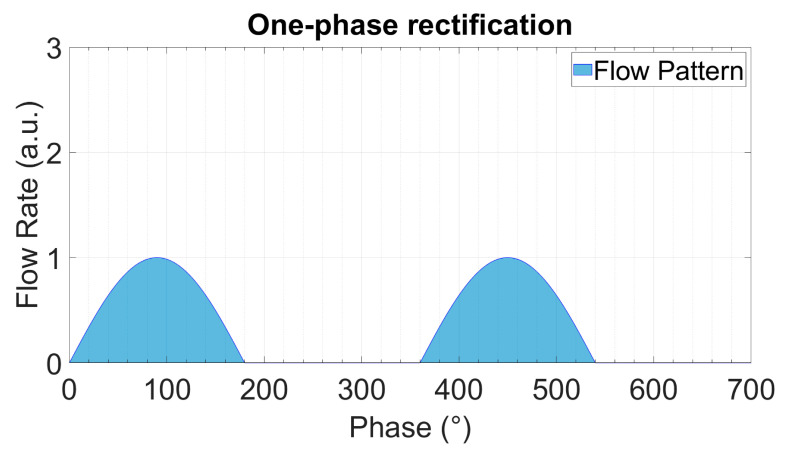	One pump. Output flow is generated by the discharge cycle of the pump, which is from 0 to 180°. The suction cycle is from 180° to 360° and does not contribute to the output flow. Because of having ideal check valves, the system has no backflow.
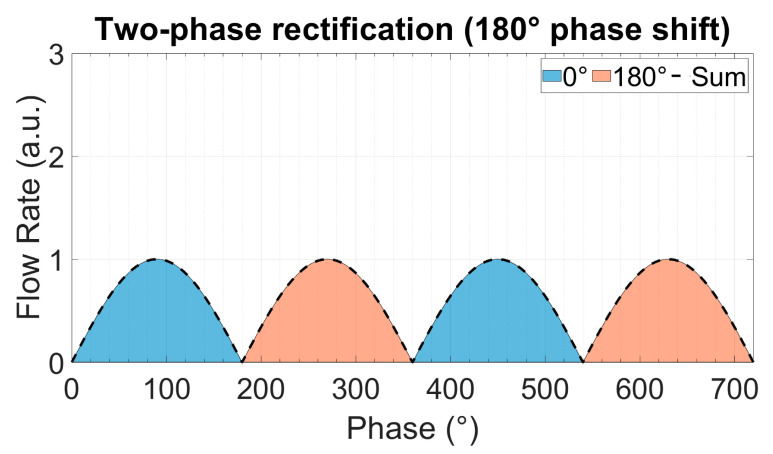	Two pumps in parallel actuating at a 180° phase difference. In addition to the first pump’s suction and discharge cycle, the second pump creates a suction and discharge cycle with a 180° phase difference, thus obtaining an effective cycloidal shape (dashed waveform) discharge flow rate, repeating itself at every 180° phase difference.
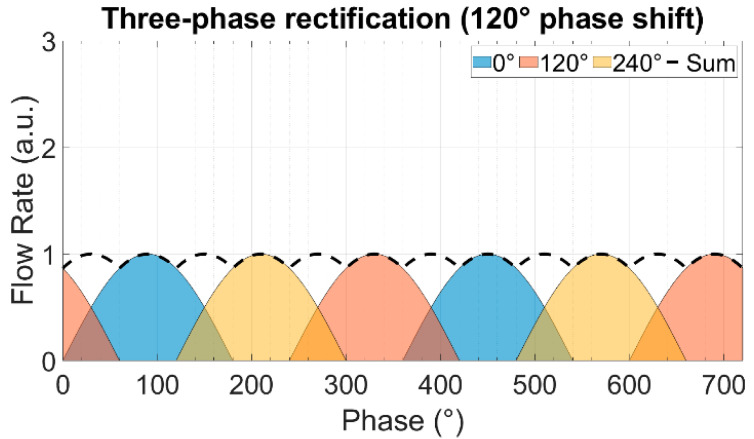	Three pumps in parallel actuating at a 120° phase difference. Effective discharge flow rate has a cycloidal shape (dashed waveform), repeating itself at every 60° phase difference, depending on the overlap of the individual fluid discharge profiles.
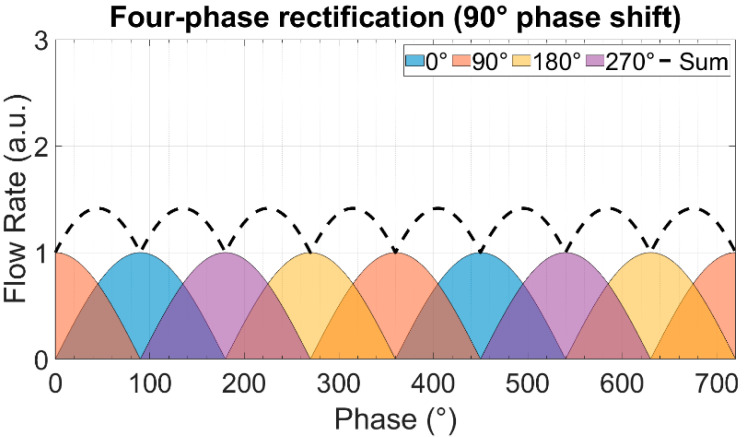	Four pumps in parallel actuating at a 90° phase difference. Effective discharge flow rate has a cycloidal shape (dashed waveform), repeating itself at every 90° phase difference.
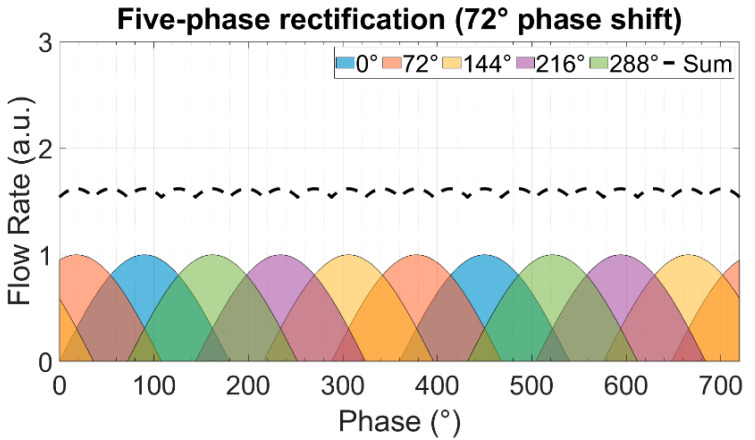	Five pumps in parallel actuating at a 72° phase difference. Effective discharge flow rate has a cycloidal shape (dashed waveform), repeating itself at every 36° phase difference.

**Table 2 sensors-23-06967-t002:** The theoretically calculated flow rate and flow ripple estimations for different multi-phase rectifiers ($Q_{stroke}$ = 1 µL/min). The values are calculated using equations 10, 11, and 12 by using the MATLAB code given in the Appendix A.

Input Parameters (Phase Rectifier)	Output Parameters (Flow)			
n (# of Pumps)	Q_DC_ (µL/min)	Q_RMS_ (µL/min)	RF_fl._	Ratio of RF_fl._ to the One-Phase Rectifier
1	0.318	0.499	1.213	100%
2	0.636	0.707	0.485	40%
3	0.955	0.956	0.042	3.4%
4	1.273	1.279	0.098	8%
5	1.591	1.592	0.015	1.2%
6	1.910	1.911	0.042	3.4%
7	2.228	2.228	0.008	0.6%
8	2.546	2.547	0.024	1.9%
9	2.865	2.865	0.005	0.4%
10	3.183	3.183	0.015	1.2%
⁞	⁞	⁞	⁞	⁞
18	5.729	5.729	0.005	0.4%
19	6.048	6.048	0.001	0.08%
20	6.366	6.366	0.004	0.3%
⁞	⁞	⁞	⁞	⁞
98	31.19	31.19	1.537 × 10^−4^	0.01%
99	31.51	31.51	3.810 × 10^−5^	0.001%
100	31.83	31.83	1.50 × 10^−4^	0.008%

**Table 3 sensors-23-06967-t003:** Comparison of the theoretical and experimental flow profiles and fluidic ripple factors obtained for 0.25 Hz of actuation. The experimentally obtained flow profile shapes deviate from the theoretical sinusoidal waveform because of the threshold pressure needed for opening the valves, fluid inertia, and the capacitance of the flexible tubing, which were not considered in the theory. Therefore, the improvement in the experimental fluidic ripple factor also deviates from the expected theoretical improvement. See Table A3 in Appendix F for further analysis.

	Flow Profile Graphs		$\mathrm{Fluidic}\;\mathrm{Ripple}\;\mathrm{Factor}\;(\sqrt{{\left(\frac{Q_{RMS}}{Q_{DC}}\right)}^{2}-1}$			
	Estimated Flow Profiles by Theoretical Study	Obtained Flow Profiles in the Experiments	Initial Model	Expected Improv.	Experimental	Obtained Improv.
One-phase	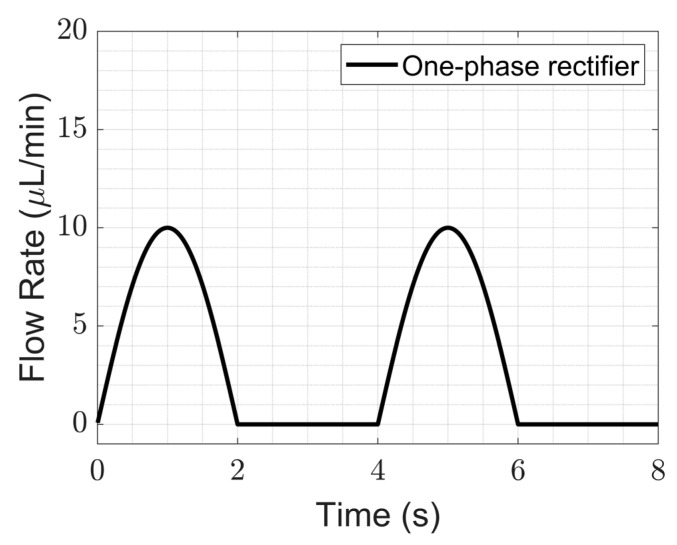	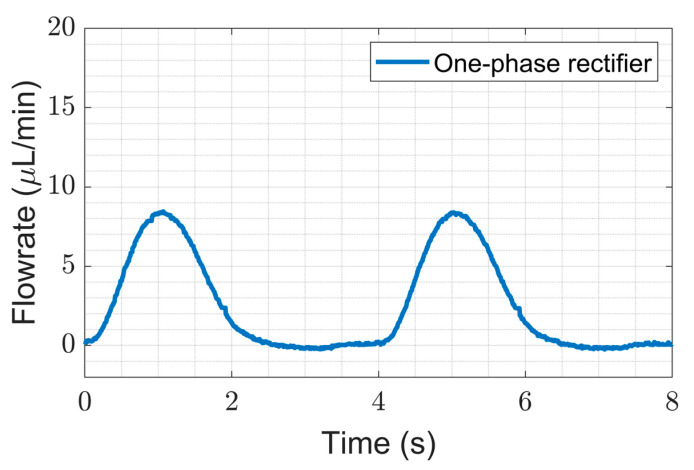	1.213	−	1.200	−
Two-phase	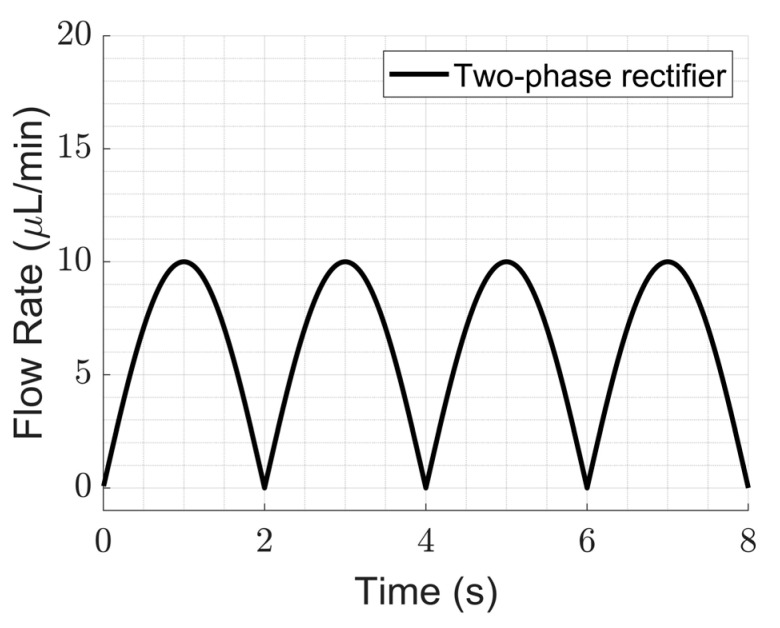	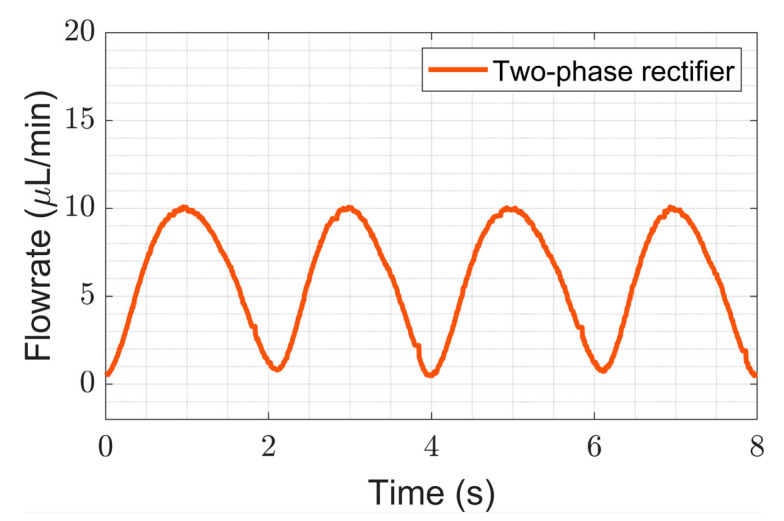	0.485	60%	0.542	55%
Three-phase	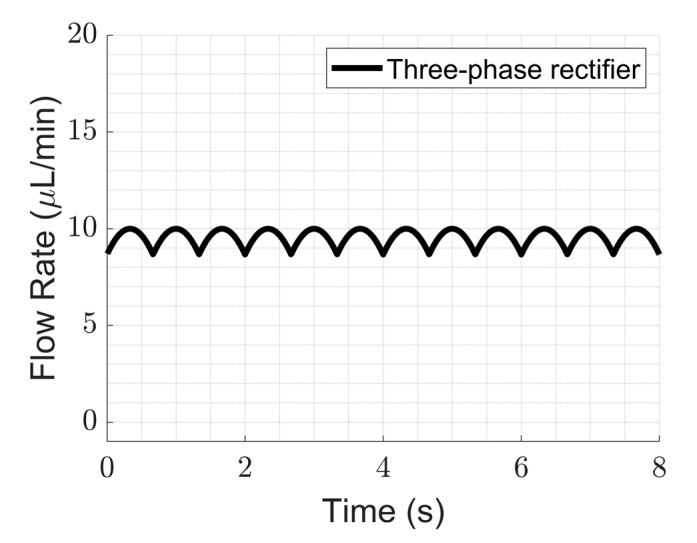	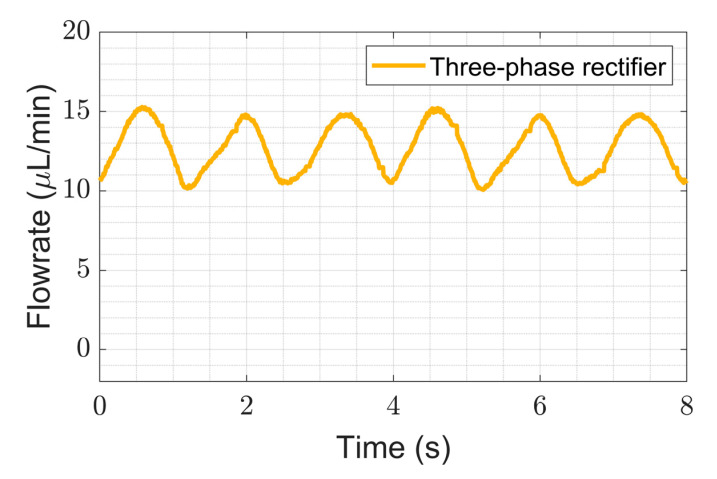	0.042	96.6%	0.126	90.5%

## Data Availability

The data that support the findings of this study are available from the corresponding authors upon request.

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
