# Peer review of "Flow Ripple Reduction in Reciprocating Pumps by Multi-Phase Rectification"

_sensors, 2023, doi:10.3390/s23156967_

Round 1

Reviewer 1 Report

This article proposes a method of flow ripple reduction in reciprocating pumps. The paper is well written and contributes multi-phase rectification scheme for minimize ripples at low flow rates, which is useful for designing a frequency-independent multi-phase fluid rectifier.

I have serval questions as follows.

 1. When each micropump is connected through the flow channel, the fluid in each flow channel will have impact in the join channel. What is about these impacts with fluidic ripple?

2.‘Every additional pump used consumes an extra voltage amplifier power.The meaning of this short sentence is not clear. Is it the disadvantage of multiple pumps in parallel?

 3.In line 360-361. it mentioned that ’A single reciprocating pump actuated at higher frequencies can also eliminate ripples in the flow but results in a high flow rate or higher power consumption.’A single reciprocating pump could work at low voltage and high frequency. The result is lower flow and power consumption. 

Reviewer 2 Report

This article presents the quantifying of the multi-phase rectification by introducing fluidic ripple factors. Overall, the authors explain their theory first and then implemented it with an experimental set-up successfully to validate. 

It can be published as it is.

Author Response

We thank the reviewer for accepting to publish our work.

Reviewer 3 Report

In this manuscript, multi-phase rectification was examined to minimize ripples for piezoelectric pumping at low flow rates through parallel connection of multiple micropumps. The potential of ripple reduction by the proposed way was analyzed theoretically and confirmed by experimental results. A fluidic ripple factor RF(fl) was used as a metric to reflect the efficiency of ripple reduction. This is a solid work with practical importance, this reviewer supports its publication in this journal and there are only a few minor issues that may need to be considered by the authors.

1.  As shown in Fig. 3 and Table 2, RF(fl) for n=5 is much better than that for n=3, it’s better to add experimental data for n=5 in section 4.2.

2. As described in lines 279-280,’The possible reason for this difference (between theoretical and experimental improvements) could be due to not having exactly identical pumps from the manufacturer.’ Is it possible to correct such difference by fine tune of the phase shift for each pump?

3. If there is a compromise among the performance, cost and complexity of the device, how many pumps should be used?

Reviewer 4 Report

Review of Flow Ripple Reduction in Reciprocating Pumps by Multi-Phase Rectification 

Comment 1. Figure 7a does not match your model in values.  It seems your model is missing important aspects of your experimental set up.  Or perhaps your phase relationships are not as accurate as desired.  

Comment 2. The 0.25hz section of figure 6 needs to be blown up and compared to your expected rectified output patterns.  I am guessing they don’t match and your probably need to include the valve behaviour if you want to get at a matching result.

Comment 3. Figure 10 is very hard to read. Show fewer cycles and an easier to follow legend.  

Comment 4. Figure 5a and Figure 6 make it quite clear that you have another time constant in play.  The capacitance (thinking electrically) in the rest of your experimental setup is not allowing your flow pulses to return to zero which they should do for the single-phase rectified system.  Your method of lowering ripple is really only useful at the lowest frequency before the time constant of your general system takes over and smooths it out anyway.  This needs significant discussion.

Comment 5.  What kind of tubing are you using and what length is it approximately.  This is in important missing piece of information.  If you are using silastic or Tygon they are going to be much more flexible than a PTFE tubing.  The more rigid the tubing the more your multi-phase system will provide benefit.  

All in all I think there is value in the work and something I had considered trying myself. Happy to be beaten to it though.  There does need to be more solid comparison of the model and reality and solid discussion of how the system damping/capacitance is affecting the result and what that implies for use cases.

Round 2

Reviewer 4 Report

The model should be compared to the experimental results in the main text of the paper, not hidden in Appendix F.

The experimental results are way too far away from the model to be acceptable.  The point of the paper was to reduce the ripple factor and your experimental results don’t show that. Your three phase which you say is the best solution has an experimental ripple in excess of 90% of the amplitude where as your model suggests it should be ~10%. You have said that you could tune the individual pumps to improve, so you should do that.

I think you need to include the diode behavior of the valves in your model and probably check on the actual flow response of your pumps or you won’t be able to match your results.  As it stands your model promises way better performance than your experimental results bear out.  Which renders your paper not publishable in its current form.  The theory is useless if it can’t be shown to match reality when applied.
